# Unlocking the Potential of Federated Learning for Deeper Models

## Abstract

Federated learning (FL) is a new paradigm for distributed machine learning that allows a global model to be trained across multiple clients without compromising their privacy. Although FL has demonstrated remarkable success in various scenarios, recent studies mainly utilize shallow and small neural networks. In our research, we discover a significant performance decline when applying the existing FL framework to deeper neural networks, even when client data are independently and identically distributed. Our further investigation shows that the decline is due to the continuous accumulation of dissimilarities among client models during the layer-by-layer back-propagation process, which we refer to as "divergence accumulation." As deeper models involve a longer chain of divergence accumulation, they tend to exhibit more significant divergence, subsequently leading to performance decline. Both theoretical derivations and empirical evidence are proposed to support the existence of divergence accumulation and its amplified effects in deeper models. To tackle this challenge, we propose a set of technical guidelines centered on minimizing divergence. These guidelines, consisting of strategies such as employing wider models and reducing the receptive field, greatly improve the performance of FL on deeper models. Their effectiveness is validated via extensive evaluation with various metrics. For example, applying the guidelines can boost the performance of ResNet101 on the Tiny-ImageNet dataset by as much as 43%.

## 1 Introduction

Federated learning (FL) is an emerging distributed learning framework that allows a global model to be trained across multiple clients and without privacy leakage McMahan et al. (2017); Li et al. (2020a); Zhang et al. (2021a). While recent FL studies achieve notable success in various contexts, they primarily utilize shallow, small-scale neural networks with typically less than ten layers Li et al. (2021b); Arivazhagan et al. (2019); Li et al. (2020b). In contrast, centralized learning (CL) often enjoys larger and deeper models due to their increased capacity for fitting a diversity of data. For example, ResNet101 He et al. (2016) has 101 layers, and Swin-L Liu et al. (2021) has 120 layers.

Naturally, the presence of this architectural disparity prompts us to question whether the performance of these deeper architectures in the FL framework aligns with that in CL. We conduct experiments on models with various depths, and unfortunately, we find that as the neural network becomes deeper, the performance gap between FL and CL often broadens significantly, even in a simple context where data across clients are independently and identically distributed (i.i.d.). According to our experiments, Fig. 1a and 1b show that the performance gap between FL and CL is noticeably more significant when using a deeper model. Additionally, Fig. 1c indicates that FL, when using a deeper model, converges more slowly and with less stability. This phenomenon hinders the utilization of FL in many applications where deeper models are necessary Anwar et al. (2018); Muhammad et al. (2018).

One straightforward approach to enhance FL performance involves making the FL process more similar to CL by reducing the local iteration number or decreasing the learning rate. This is based on the fact that when the number of local iterations in FL is set to one, it becomes equivalent to CL. However, these methods contradict the core motivation of FL, which not only impacts the communication efficiency of FL but also raises concerns about privacy protection Hu et al. (2020).

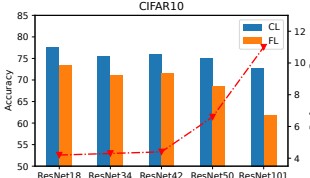 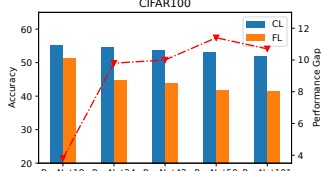 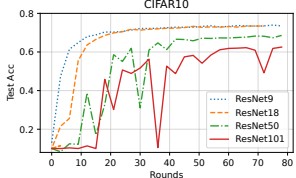

(a) Comparison between CL and FL on CIFAR10.

(b) Comparison between CL and FL on CIFAR100.

(c) Training curves for FL on CI-FAR10 with different models.

Figure 1: This experiment is conducted on various datasets using ResNets He et al. (2016) with different depths. We choose 30 clients, 8 local epochs and learning rate to be 0.02. The dashed line in Fig. 1a, 1b show the performance gap between FL and CL.

To tackle this issue and attain improved training performance with deeper models in FL, we must address a critical question: *Why does the utilization of deeper networks in FL result in a more pronounced performance decline when compared to using shallower networks?*

One conjecture suggests that deeper models with a larger number of parameters are more prone to overfitting the local data on each client, resulting in a degradation of overall performance. However, our experiments cast doubt on this conjecture. If it were true, increasing the number of model parameters, either by widening or deepening the model, should consistently lead to a decline in performance. Surprisingly, when we increase the number of model parameters by widening each layer, we observe an improvement in FL performance instead of a decline. This indicates that attributing the performance degradation solely to the number of parameters is insufficient. For more detailed information on this experiment, refer to Section 4.

To provide a clearer understanding of the aforementioned question, we introduce the concept of *divergence* for client models in FL. Divergence, in this context, refers to the dissimilarity between the update directions of each client and serves as a significant indicator for evaluating FL performance. We demonstrate that the divergence is zero if, and only if, the optimization objective of FL aligns with that of CL. A detailed proof is provided in Section 3. Thus, a lower divergence implies a closer approximation of FL model to CL model, which further indicates enhanced performance of this FL model. Formally, we define divergence as follows. For training round $t$ and model layer $L$, the average divergence $div_t^L$ of all clients is formally defined as:

$$div_t^L = \frac{1}{N}\sum_{i=1}^{N}\sqrt{\frac{||\mathbb{E}_{x\in X_i}[\nabla_\theta L(\theta_t^L, x)] - \nabla\bar{\theta}_t^L||_2^2}{d^L}}, \quad \nabla\bar{\theta}_t^L = \frac{1}{N}\sum_{i=1}^{N}\mathbb{E}_{x\in X_i}[\nabla_\theta L(\theta_t^L, x)] \quad (1)$$

where $\theta_t^L \in \mathbb{R}^{d_L}$ is the global model parameter with dimension $d^L$, $N$ is the number of clients, $X_i$ is the dataset of client $i$, $\mathbb{E}_{x\in X_i}[\nabla_\theta L(\theta_t^L, x)]$ is the expected gradient calculated in client $i$ and $\nabla\bar{\theta}_t^L$ is the averaged expectation across all clients.

We demonstrate the divergences of ResNet101 trained on Tiny-ImageNet with client number to be 30, and the results are shown in Fig. 2a. Here are several noteworthy observations derived from this figure. Firstly, it is evident that compared with deeper layers, the divergence for shallower layers is usually larger. Secondly, divergences for various layers exhibit distinct properties. In the case of shallow layers, divergences generally decrease and eventually converge. However, for deep layers, the divergences tend to increase and intensify.

These findings emphasize a strong relationship between parameter divergence and model architecture. To shed light on this phenomenon, we introduce a theorem centered around divergence accumulation. The fundamental idea is illustrated in Fig. 2b. In short, the divergence of each layer is influenced by the back-propagation algorithm and can be broken down into two components: *the divergence back-propagated from the subsequent layer* and *the divergence originating from distinct inputs*. As each layer integrates divergence from the subsequent layer, divergences accumulate at each step of the back-propagation process. Through this layer-by-layer progression, the divergences of later layers steadily accumulate upon the divergences of preceding layers. Consequently, *deeper networks exhibit longer chains of divergence accumulation*, ultimately resulting in lower performance.

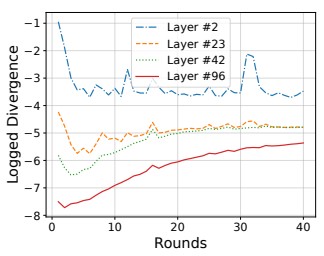 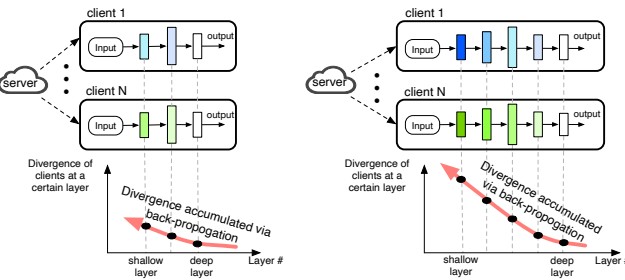

(a) Divergences in ResNet101. For better visualization we use $\log(\cdot)$ to rescale these divergences.

(b) Divergence accumulation in a shallow network (left) and deeper network (right). As deeper network has a longer accumulation chain, it tends to present a larger divergence in shallower layers.

Figure 2: Illustrations about divergence accumulation.

Building upon the aforementioned theorem, we propose several guidelines to enhance the training of deeper networks in the FL framework, focusing on two key aspects. The first aspect relates to *model architecture*. We have discovered that using wider models and reducing receptive fields can mitigate the adverse effects of divergence accumulation and therefore significantly improve FL performance. The second aspect concerns the *pre-processing of input data*. As the fundamental cause of model divergence is the diversity of data originating from different clients, implementing appropriate pre-processing strategies, including using images with higher resolution and adopting proper data augmentation methods, can mitigate the divergence at the root level, thus substantially enhancing the performance of deeper models in FL.

It is crucial to emphasize that our focus is not on developing "new" methods for enhancing model performance. Instead, we draw attention to existing techniques that are already utilized in CL. Nevertheless, our experiments reveal that when applied in the context of FL, these methods have a remarkable impact in reducing divergence, resulting in significantly greater enhancements in FL performance compared to CL. More importantly, we observe that these methods can largely reduce the performance gap between FL and CL. By highlighting these guidelines, our objective is to enhance the understanding of the key considerations in FL as compared to CL. In summary, the contributions of this paper are the following:

First, we observe a noteworthy phenomenon: deeper models in the context of FL often face challenges in achieving convergence, resulting in a degradation of performance. We consider this issue to be critical, especially in the implementation of large-scale FL systems. To the best of our knowledge, our paper is the first to systematically investigate this phenomenon, exploring its underlying causes and proposing potential solutions.

Second, we introduce the concept of "divergence" as a metric to quantify the dissimilarity in update directions across clients in FL. Utilizing this measurement, we make an important observation: divergences for shallow layers are usually larger then that in deeper layers. This finding sheds light on the behavior and performance characteristics of different layers in FL models.

Third, to offer a rationale for the aforementioned observation, we present a theorem that centered on divergence accumulation. To gain a thorough understanding of the process of divergence accumulation in the context of FL, we provide both experimental evidence and theoretical analysis.

Finally, we propose several principles that serve as guidelines to reduce divergence and enhance the performance of FL, particularly in the context of deeper models. We hope that by adhering to these principles, practitioners can enhance the performance of FL when employing deeper models.

## 2    RELATED WORK

**Federated Learning with deeper neural networks.** There are a few works applying deeper models to FL. Some works utilize pretrained large models He et al. (2021); Luo et al. (2022); Gong et al. (2021) and their goal is using FL to finetune on various datasets. This setting is different from ours

that trains a global model collaboratively from scratch. Some other works Zhang et al. (2021b); Panda et al. (2022) train deeper models with high-resolution images (e.g. CT images with resolution of $1024 \times 1024$). Our analysis shows that image with high resolution is beneficial for deeper models in FL. In summary, although there are currently some works using deeper models and achieving considerable results in different FL scenarios, they either require some specific conditions (such as using pretrained models) or need to satisfy certain properties (such as high-resolution images). To the best of our knowledge, there is no previous work that systematically analyzes the causes and solutions for the difficulties in applying deeper models to FL.

**Neural Architecture Search (NAS) for Federated Learning.** The goal of NAS is to find the optimal model architecture for a specific task. In the field of FL, NAS needs to consider not only model performance, but also additional factors, such as the computational capabilities of edge devices and communication overhead Yuan et al. (2020); Zhu et al. (2021); Zhu & Jin (2021); Wang et al. (2023). However, due to the large search space of deep networks, most of the research focuses on shallow network design, leading to a lack of studies on deeper network structures.

## 3 PHENOMENON OF DIVERGENCE ACCUMULATION

### 3.1 DIVERGENCE

To initiate our analysis, we will first elaborate on the concept of divergence. In essence, the intuition of divergence is to measure the distinctions in optimization objectives between FL and CL. For CL, the optimal model parameters $\theta^*$ satisfies:

$$\mathbb{E}_{x \in X}[\nabla_{\theta^*} L(\theta^*, x)] = 0, \tag{2}$$

where $X$ is the dataset and $L(\cdot)$ is the loss function. In FL, on the other hand, the optimal global model parameter $\theta^*$ satisfies:

$$\sum_{i=1}^{N} \mathbb{E}_{x \in X_i}[\nabla_{\theta^*} L(\theta^*, x)] = 0, \tag{3}$$

where $X_i$ is dataset of the i'th client. Assume datasets across clients are i.i.d, we can derive that the optimal solution of FL equals to that of CL when and only when:

$$\mathbb{E}_{x \in X_i}[\nabla_{\theta^*} L(\theta^*, x)] = 0, \forall i \in 1, \cdots, N \tag{4}$$

Under this circumstance, recall the divergence defined in Eq. 1, we have:

$$div = \frac{1}{N} \sum_{i=1}^{N} \sqrt{\frac{||\mathbb{E}_{x \in X_i}[\nabla_{\theta^*} L(\theta^*, x)] - \nabla \bar{\theta}_t||_2^2}{d}} = 0. \tag{5}$$

In essence, the optimization objective for FL aligns with that of CL only when the divergence reaches zero. This deduction provides us with a crucial insight: when there exists substantial divergence among the client models, FL might result in significant performance degradation compared to CL. This forms the basis for our introduction of the concept of divergence.

### 3.2 PHENOMENON OF DIVERGENCE ACCUMULATION

In this section, we introduce a theorem based on the previously defined divergence, which provides insights into the causes of divergence accumulation in FL. Our objective is to establish a connection between the diversity of data and the divergence of model. Formally, we aim to prove that the expected divergence is accumulated during the back-propagation process, that is, the expectation of divergence in a shallow layer is always greater than that in a deep layer.

To begin our analysis, we examine the data. Considering that data across different clients can be seen as random variables, we assume their distributions are identical. For instance, in a classification task, we can represent the data in client $k$ belonging to class $c$ as follows:

$$\mathbf{X_k^c} = \bar{\mathbf{X}}^c + \tilde{\mathbf{X}}_k^c \text{ ,where } \mathbb{E}_{x \in \bar{\mathbf{X}}^c}[\nabla_\theta L(\theta, x)] = \mathbb{E}_{x \in \mathbf{X_k^c}}[\nabla_\theta L(\theta, x)] \text{ and } \tilde{\mathbf{X}}_k^c \sim p(\tilde{\mathbf{X}}|c). \tag{6}$$

In the given equation, $\tilde{\mathbf{X}}_k^c$ represents a random variable that is associated with class $c$ and reflects the diversity of data in the $k$-th client. On the other hand, $\bar{\mathbf{X}}^c$ is also associated with class $c$ but represents

the generality or typicality of that class. The gradient calculated using $\bar{\mathbf{X}}^{\mathbf{c}}$ is equivalent to using the original dataset $\mathbf{X}_{\mathbf{k}}^{\mathbf{c}}$. It can be understood as a prototype concept similar to what has been introduced in previous literature Yang et al. (2018); Tan et al. (2022); Xu et al. (2020).

We next consider the model. To illustrate, we will consider an example involving two linear layers. The forward calculation process for layers $i$-1 and $i$ can be described as follows:

$$\mathbf{H_{i-1}} = \mathbf{A_{i-1}} \cdot \mathbf{Z_{i-2}} + \mathbf{b_{i-1}}, \quad \mathbf{Z_{i-1}} = \sigma(\mathbf{H_{i-1}}), \quad \mathbf{H_i} = \mathbf{A_i} \cdot \mathbf{Z_{i-1}} + \mathbf{b_i}, \tag{7}$$

where $\mathbf{A}$, $\mathbf{b}$ denotes model parameters and $\mathbf{Z}$, $\mathbf{H}$ denotes intermediate calculation results. Assume the loss function is $L(\cdot)$, according to the chain rule of derivation, we have:

$$\frac{\partial L}{\partial \mathbf{A_i}} = \frac{\partial L}{\partial \mathbf{H_i}} \frac{\partial \mathbf{H_i}}{\partial \mathbf{A_i}} = \frac{\partial L}{\partial \mathbf{H_i}} \mathbf{Z_{i-1}}^T, \frac{\partial L}{\partial \mathbf{Z_{i-1}}} = \frac{\partial L}{\partial \mathbf{H_i}} \frac{\partial \mathbf{H_i}}{\partial \mathbf{Z_{i-1}}} = \mathbf{A_i^T} \frac{\partial L}{\partial \mathbf{H_i}} \tag{8}$$

$$\frac{\partial L}{\partial \mathbf{H_{i-1}}} = \frac{\partial L}{\partial \mathbf{Z_{i-1}}} \frac{\partial \mathbf{Z_{i-1}}}{\partial \mathbf{H_{i-1}}} = \mathbf{A_i^T} \frac{\partial L}{\partial \mathbf{H_i}} \odot \sigma'(\mathbf{H_{i-1}}), \tag{9}$$

where $\odot$ denotes the element-wise product. Based on these equations, we can derive the relationship between the gradients in adjacent layers:

$$\frac{\partial L}{\partial \mathbf{A_{i-1}}} = \mathbf{A_i^T}(\frac{\partial L}{\partial \mathbf{A_i}}(\mathbf{Z_{i-1}^T})^{-1} \odot \sigma'(\mathbf{H_{i-1}}))\mathbf{Z_{i-2}}^T. \tag{10}$$

Define : $\epsilon_{\mathbf{i}} = \frac{\partial L}{\partial \mathbf{A_i}} - \frac{\bar{\partial} L}{\partial \mathbf{A_i}}$, where $\frac{\bar{\partial} L}{\partial \mathbf{A_i}}$ is the gradient calculated by the prototype $\bar{X}$. According to the definition for $\epsilon_i$, we have $\mathbb{E}[\epsilon_{\mathbf{i}}] = \mathbf{0}$ and the divergence for model in layer $i$ is $\mathbb{E}[||\epsilon_{\mathbf{i}}||_2]$ according to Eq. 1. Rewrite Eq. 10 using this definition, we have:

$$\frac{\partial L}{\partial \mathbf{A_{i-1}}} = \mathbf{A_i^T}(\epsilon_i(\mathbf{Z_{i-1}^T})^{-1} \odot \sigma'(\mathbf{H_{i-1}}))\mathbf{Z_{i-2}^T} + \mathbf{A_i^T}(\frac{\bar{\partial} L}{\partial \mathbf{A_i}}(\mathbf{Z_{i-1}^T})^{-1} \odot \sigma'(\mathbf{H_{i-1}}))\mathbf{Z_{i-2}^T}. \tag{11}$$

Now, we examine the phenomenon of divergence accumulation, which serves as the foundation of this study. To facilitate our analysis, we introduce two mild assumptions.

*Assumption 1:* The expectation of divergence is retained during the back-propagation process, that is: $\mathbb{E}[||\mathbf{A_i^T}(\epsilon_i(\mathbf{Z_{i-1}^T})^{-1} \odot \sigma'(\mathbf{H_{i-1}}))\mathbf{Z_{i-2}}^T||^2] = \mathbb{E}[||\epsilon_{\mathbf{i}}||^2]$.

Assumption 1 can be easily satisfied because there exist sophisticated techniques in the training of neural networks to maintain the gradient norm and address issues like gradient explosion or vanishing. These techniques include parameter initialization Glorot & Bengio (2010); He et al. (2015), residual links He et al. (2016), and normalization Ioffe & Szegedy (2015); Ba et al. (2016). Consequently, the back-propagated divergence can also be retained.

*Assumption 2:* The random variable vector $\epsilon_{\mathbf{i}}$ is independent on $\mathbf{Z}$ and $\mathbf{H}$.

*Theorem 1:* Given Assumption 1 and Assumption 2, it follows that the divergence of any given layer is no smaller than that of the subsequent layer. Formally, this can be expressed as: $\mathbb{E}[||\epsilon_{\mathbf{i-1}}||^2)] \geq \mathbb{E}[||\epsilon_{\mathbf{i}}||^2]$.

The proof is outlined as follows: the expected divergence of layer $i$-1 is comprised of two components. The first is the divergence back-propagated from layer $i$ and the second is the divergence arising from distinct inputs $\mathbf{Z}$ and $\mathbf{H}$. As a result, the divergence will consistently increase during back-propagation. Notably, we demonstrate that similar proofs can be employed to extend this theorem to other model structures, such as Convolutional Neural Networks (CNNs).

The above theorem connects data diversity and model divergence in FL. Through the layer-by-layer back-propagation process, the divergences of subsequent layers accumulate on top of the divergences of previous layers. This accumulation is more pronounced in deeper networks due to their longer chains of divergence. Consequently, deeper networks tend to exhibit higher divergences, resulting in a decline in performance. This analysis provides an explanation for the observations presented in Section 1. Let us recapitulate the observations along with their corresponding explanations.

**Observation 1:** Deeper models are challenging to converge. **Explanation:** In deeper models, gradients need to be calculated through a long chain of back-propagation. The divergence accumulates at each step of back-propagation and leads to a significant divergence in the final gradient, ultimately causing a decline in the performance of the model.

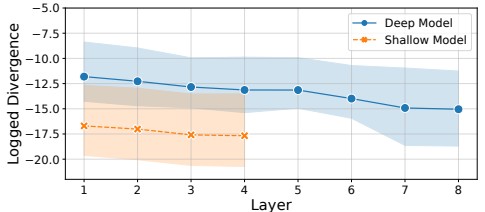 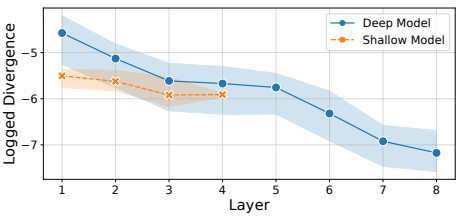

(a) Logged divergences for data with Gaussian noises.  (b) Logged divergences for data in the same class.

Figure 3: In this experiment, we demonstrate the divergences of a 8-layer CNN (deep model) and a 4-layer CNN (shallow model) trained on MNIST. We use the log function $\log(\cdot)$ to transform divergences. The dark-colored lines represent the mean of divergence within different classes, and the light-colored areas represent the range of real divergences.

**Observation 2:** Shallow layers tend to converge, while deep layers tend to diverge. **Explanation:** Referring to Eq. 10, the divergence for layer $i$ is influenced by its input $\mathbf{Z_{i-2}}$. If the previous layers do not converge, the inputs $\mathbf{Z_{i-2}}$ in different clients are computed using different previous layers and different data, leading to significant divergences. However, when the previous layers do converge, indicating high similarity among their preceding layers, the inputs $\mathbf{Z_{i-2}}$ have lower divergences. This also explains our earlier observation that the latter layers do not converge before the former ones.

### 3.3 Empirical Validation of Divergence Accumulation

We conduct a series of experiments to validate the phenomenon of divergence accumulation discussed in the preceding subsection. Our goal is to confirm the existence of the following two phenomena: 1) The accumulation of divergence during the back-propagation process, and 2) The greater divergence in earlier layers of deeper models resulting from their longer accumulation chain. Two approaches are employed to reflect the diversity of data across different clients.

Firstly, we utilize images with Gaussian noises to model data diversity. We begin by calculating the gradient using an original image. Then, we add random Gaussian noises (drawn from $\mathcal{N}(0, 0.1)$) to the input image and calculate the corresponding gradients. We sample the noises 1000 times, and the standard deviation among these gradients precisely represents the model divergence when updating the same model with different image data.

Secondly, we use images from the same class to model data diversity. We individually calculate the gradients for all images belonging to the same class. Similarly, we use the standard deviation among these gradients to reflect the model divergence resulting from different image training data.

In our experiments, we employ two types of models, namely shallow and deep models, for calculations. To ensure the robustness of the results, each experiment is repeated three times using different random seeds, and we provide the confidence interval in each figure.

Divergences under the two types of data diversities are plotted layer-wisely in Fig. 3. The plots clearly illustrate that, for both types of data diversity, the divergence within the same model accumulates during the back-propagation process. When comparing the shallow model and the deep model, it is evident that the shallow model exhibits lower divergence in preceding layers due to its shorter accumulation chain. The findings suggest that the phenomenon of divergence accumulation holds true in typical and realistic scenarios.

## 4 Enhancing the Performance of Deeper Neural Networks in FL

In this section, we put forth a set of guidelines to enhance the performance of FL when utilizing deeper neural networks. As the divergence of models plays a crucial role in determining the effectiveness of FL, the central goal is therefore to minimize these divergences during the FL training process.

Table 1: Experiments conducted using a 13-layer ResNet with different widths. Acc. is the best model accuracy at test time and Div. is the mean divergence of the model of all layers.

| Dataset | Method | 1×width | | | 2×width | | | 3×width | | |
|---|---|---|---|---|---|---|---|---|---|---|
| | | Acc. | Div. | Gap | Acc. | Div. | Gap | Acc. | Div. | Gap |
| Tiny-Imagenet | CL | 0.509 | - | 0.051 | 0.526 | - | 0.042 | 0.546 | - | 0.041 |
| | FL | 0.458 | 0.046 | | 0.484 | 0.024 | | 0.504 | 0.011 | |
| CIFAR100 | CL | 0.587 | - | 0.060 | 0.623 | - | 0.027 | 0.634 | - | 0.021 |
| | FL | 0.527 | 0.052 | | 0.596 | 0.032 | | 0.613 | 0.014 | |

As shown in Eq. 11, the two terms on the right hand side of the equation represent the dual factors influencing the divergence for a specific layer: the divergence back-propagated from the subsequent layer and the divergence originating from distinct inputs. These factors contribute independently to the overall divergence. The first component, related to the model architecture, plays a significant role in influencing the divergence. We will demonstrate that specific architectural designs, such as utilizing wider models with smaller receptive fields, can effectively reduce model divergences. The second component is associated with the input data. We will illustrate that by increasing the similarity of data across different clients, we can correspondingly reduce the model divergences. Augmenting the similarities of data among clients in FL leads to a decrease in divergences, thereby significantly enhancing the performance of FL when utilizing deeper models.

To effectively evaluate the performance of various designs, we employ two three metrics: Test Accuracy (denoted as **Acc.**), Mean Divergence (denoted as **Div.**) and Performance Gap between CL and FL(denoted as **Gap**). As our primary objective in this experiment is to address the challenges faced in FL training when compared to CL, we are especially interested in the performance gap between these two methods rather than the absolute performances of FL.

In the following subsections, we will introduce guidelines and present corresponding empirical evidence. First, we introduce the general experimental settings. We use the Tiny-ImageNet dataset tin (2015) (10,000 training images, 1,000 test images), the CIFAR100 dataset Krizhevsky et al. (2009) (50,000 training images, 10,000 test images), and the CIFAR10 dataset Krizhevsky et al. (2010) (50,000 training images, 10,000 test images). For the default data preprocessing, we resize all images to a size of 64×64 and applied Random Resized Crop (RRC) as data augmentation on the training set. We set the number of clients to 30, the number of local training epochs to 8, and the learning rate to 0.02. We utilize FedAvg as the baseline algorithm as the data on each client are i.i.d. In terms of the construction of experimental platform, we simulate multiple clients with a single A100 GPU and conducted simulation tests on it.

### 4.1 GUIDELINES FOR ENHANCING MODEL ARCHITECTURES

In this subsection, we propose several guidelines to decrease model divergence across various clients. As the accumulation of divergence is associated with back-propagation and consequently linked to the model's architecture, we can reduce divergence by strategically designing the model architecture.

**Using wider models.** In CL, prior research has demonstrated the benefits of utilizing "wider" network models. In the context of FL, our experimental findings, as presented in Table 1, confirm that wider neural networks yield greater enhancements compared to CL. Notably, we observe a reduction in divergence among clients when employing wider networks. This phenomenon can be attributed to the "lazy" characteristic exhibited by wider neural networks, as suggested by previous studies Chizat et al. (2019). The reduced parameter changes within each network contribute to a decrease in divergence among models across different clients. Consequently, this reduction in divergence fosters improved FL performance, aligning with our overarching objective.

**Using models with smaller receptive fields.** The receptive field refers to the region of the input image that influences the activation of a particular neuron and is important in CNNs. Neurons with smaller receptive fields predominantly extract low-level semantic information, while those with larger receptive fields focus on high-level semantic information. The receptive field is calculated using the recursive formula $l_k = l_{k-1} + [(K_k - 1) \prod_{i=1}^{k-1} s_i]$, where $l_k$ represents the receptive field of the $k$-th layer, $K_k$ and $s_i$ denote the kernel size and stride of the $k$-th layer, respectively.

Table 2: Experiments conducted using ResNet101 with different receptive fields. We tried two settings: replacing the first 7×7 convolution with a 3×3 one (**3×3 Conv.**), and removing the MaxPooling layer (**No M.P.**). Both settings reduce the receptive field of subsequent convolutional layers.

| Dataset | Method | Base | | | 3×3 Conv. | | | No M.P. | | |
|---|---|---|---|---|---|---|---|---|---|---|
| | | Acc. | Div. | Gap | Acc. | Div. | Gap | Acc. | Div. | Gap |
| Tiny-Imagenet | CL | 0.340 | - | 0.104 | 0.472 | - | 0.015 | 0.486 | - | 0.013 |
| | FL | 0.236 | 0.012 | | 0.457 | 0.008 | | 0.473 | 0.008 | |
| CIFAR100 | CL | 0.510 | - | 0.097 | 0.594 | - | 0.02 | 0.602 | - | 0.059 |
| | FL | 0.413 | 0.014 | | 0.574 | 0.011 | | 0.543 | 0.012 | |
| CIFAR10 | CL | 0.728 | - | 0.11 | 0.775 | - | 0.074 | 0.793 | - | 0.069 |
| | FL | 0.618 | 0.014 | | 0.701 | 0.012 | | 0.724 | 0.010 | |

Table 3: Experiments conducted using ViT and Swin with 6 transformer blocks. For ViT, we choose patch size to be 4 with dimension 512. For Swin the patch size is 4 with window size 4. The depth and parameter number of these models are similar and compatible to ResNet18. Numbers in brackets refer to the ratio of performance degradation.

| Model | Tiny-Imagenet | | CIFAR100 | | CIFAR10 | |
|---|---|---|---|---|---|---|
| | CL | FL | CL | FL | CL | FL |
| ViT | 0.340 | 0.236 (↓30.5%) | 0.572 | 0.454 (↓20.6%) | 0.684 | 0.572 (↓15.8%) |
| Swin | 0.389 | 0.326 (↓16.2%) | 0.499 | 0.433 (↓13.2%) | 0.663 | 0.607 (↓8.45%) |

To explore the impact of receptive field size on FL performance, we conducted experiments with two settings aimed at reducing the receptive fields. The results, presented in Table 2, reveal that minimizing the receptive field effectively limits the divergence between models, thereby enhancing the overall FL performance. This observation is attributed to the fact that a smaller receptive field ensures that neurons primarily observe similar image regions, such as low-level semantic information like edges and corners. By reducing the diversity of input data, we effectively reduce model divergence, as indicated by our earlier analysis.

Additionally, we assessed the performance of two vision transformer models, ViT Dosovitskiy et al. (2020) and Swin Liu et al. (2021), which offer contrasting approaches to receptive field design. While ViT boasts infinitely large receptive fields for neurons in each layer, Swin maintains a specific receptive field size. We conducted experiments using both models, each comprising six transformer blocks and a parameter count similar to ResNet18. The experimental results, presented in Table 3, demonstrate that the Swin transformer exhibits a relatively smaller performance decline in FL compared to the ViT model. This finding further validates our guideline of utilizing models with small receptive fields.

## 4.2 GUIDELINES FOR OPTIMIZING DATA PRE-PROCESSING

In this subsection, we put forth guidelines focused on data-centric approaches. The primary driver of model divergence stems from the diversity of data sourced from various clients. By implementing suitable pre-processing strategies for data, this divergence can be greatly decreased.

**Using image with higher resolution if possible.** We note that adjusting the image resolution and modifying the model's receptive field are two distinct implementation methods that share a common underlying principle. Reducing the receptive field enables individual neurons to concentrate on smaller pixel regions, whereas increasing the image resolution leads to less image information within the same-sized pixel area. Both methods fundamentally serve a similar purpose. The findings presented in Table 4 align with our expectations, showcasing that FL models yield improved performance when working with higher-resolution images.

**Using proper data augmentation methods.** By adopting appropriate data augmentation methods, it is possible to enhance the similarity of data within each client, leading to a reduction in model divergences. In our experiments, we employed two popular techniques: Random-Resized-Crop (RRC)

Table 4: Experiments conducted using ResNet101 with different image resolution. We select the image sizes to be **64×64**, **128×128**, **196×196**, respectively.

| Dataset | Method | 64×64 | | | 128×128 | | | 192×192 | | |
|---|---|---|---|---|---|---|---|---|---|---|
| | | Acc. | Div. | Gap | Acc. | Div. | Gap | Acc. | Div. | Gap |
| Tiny-Imagenet | CL | 0.340 | - | 0.104 | 0.495 | - | 0.035 | 0.540 | - | 0.05 |
| | FL | 0.236 | 0.012 | | 0.460 | 0.008 | | 0.490 | 0.006 | |
| CIFAR100 | CL | 0.510 | - | 0.097 | 0.604 | - | 0.103 | 0.645 | - | 0.042 |
| | FL | 0.413 | 0.014 | | 0.501 | 0.012 | | 0.603 | 0.010 | |
| CIFAR10 | CL | 0.728 | - | 0.11 | 0.785 | - | 0.088 | 0.799 | - | 0.045 |
| | FL | 0.618 | 0.014 | | 0.697 | 0.012 | | 0.754 | 0.010 | |

and Color-Jitter (CJ). RRC involves randomly cropping a region from an image and subsequently resizing it back to its original dimensions. On the other hand, CJ refers to randomly altering the color attributes of the image. The experimental results presented in Table 5 underscore the benefits of employing suitable data augmentation techniques. However, excessive or inappropriate augmentation approaches may lead to a slowdown in local training speed and result in a degradation of model accuracy.

Table 5: Experiments on ResNet101 with 3 augmentation methods: Random Resized Crop (**RRC**), Color Jitter (**CJ**) and mixture of RRC and CJ (**Both**).

| Dataset | Method | RRC | | | CJ | | | Both | | |
|---|---|---|---|---|---|---|---|---|---|---|
| | | Acc. | Div. | Gap | Acc. | Div. | Gap | Acc. | Div. | Gap |
| Tiny ImageNet | CL | 0.340 | - | 0.104 | 0.218 | - | 0.071 | 0.321 | - | 0.069 |
| | FL | 0.236 | 0.012 | | 0.147 | 0.008 | | 0.252 | 0.008 | |
| CIFAR100 | CL | 0.510 | - | 0.097 | 0.330 | - | 0.095 | 0.455 | - | 0.105 |
| | FL | 0.413 | 0.014 | | 0.235 | 0.016 | | 0.350 | 0.014 | |

### 4.3 DISCUSSION

As revealed in our results, the proposed enhancement methods also have a positive impact on CL. However, the improvement in FL is notably more significant, and the performance gap between FL and CL has diminished. This suggests that the enhancements in FL are inherently linked to the unique aspects of the distributed training process, which we believe are associated with divergence.

For a deeper understanding, let us denote the improvement in FL as $I_{total} = FL_2 - FL_1$, where $FL_2$ and $FL_1$ represent FL performance before and after applying our proposed methods, respectively. We can decompose $I_{total}$ into $I_{total} = I_{cent} + I_{div}$. Here, $I_{cent}$ represents the improvement similar to that in CL, while $I_{div}$ signifies the enhancement resulting from a reduction in divergence. This former improvement, $I_{cent}$, can be quantified as $I_{cent} = CL_2 - CL_1$, where $CL_2$ and $CL_1$ denote the performance metrics of CL before and after our enhancements. Therefore, $I_{div} = (CL_1 - FL_1) - (CL_2 - FL_2)$. The first term on the right-hand side captures the performance gap between CL and FL before the introduction of our enhancements, while the latter term represents the gap after implementation. Based on our experimental observations, it's evident that $CL_1 - FL_1$ exceeds $CL_2 - FL_2$. This suggests that the enhancement in FL performance is not solely attributable to factors that also affect CL; rather, it is closely related to the reduction in model divergence after applying our proposed methods.

## 5 CONCLUSION

In this paper, we observe that deeper neural networks are difficult to converge in FL, which we believe is a critical problem for large-scale FL. To gain a deeper understanding, we introduce and examine the phenomenon of divergence accumulation. Finally, several guidelines are proposed to reduce the divergence, which greatly improve"[;] the performance of FL on deeper models. We believe that this work holds significant value and serves as a source of inspiration for future research in large-scale FL.

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

## 6 APPENDIX

### 6.1 PROOF FOR DIVERGENCE ACCUMULATION

In this section, we provide a detailed proof of *Theorem 1* proposed in Section 3. Let us review the content and relevant assumptions of the theorem first:

*Assumption 1:* The expectation of divergence is retained during the back-propagation process, that is:

$$\mathbb{E}[||\mathbf{A_i^T}(\epsilon_\mathbf{i}(\mathbf{Z_{i-1}^T})^{-1} \odot \sigma'(\mathbf{H_{i-1}}))\mathbf{Z_{i-2}}^T||^2] = \mathbb{E}[||\epsilon_\mathbf{i}||^2] \tag{12}$$

*Assumption 2:* The random variable matrix $\epsilon_\mathbf{i}$ is independent with $\mathbf{Z}$ and $\mathbf{H}$.

*Theorem 1:* Given Assumption 1 and Assumption 2, it follows that the divergence of previous layer is no smaller than that of the later layer. Formally, this can be expressed as:

$$\mathbb{E}[||\epsilon_\mathbf{i-1}||^2)] \geq \mathbb{E}[|\epsilon_\mathbf{i}||^2] \tag{13}$$

*Proof:* According to the definition of $\epsilon_\mathbf{i-1}$, we have:

$$||\epsilon_\mathbf{i-1}||^2 = ||\frac{\partial L}{\partial \mathbf{A_{i-1}}} - \frac{\bar{\partial L}}{\partial \mathbf{A_{i-1}}}||^2 \tag{14}$$

Using Eq. 10 to rewrite this equation:

$$||\epsilon_\mathbf{i-1}||^2 = ||\mathbf{A_i^T}(\epsilon_i \mathbf{Z_{i-1}^{-1}} \odot \sigma'(\mathbf{H_{i-1}}))\mathbf{Z_{i-2}} + \mathbf{A_i^T}(\frac{\bar{\partial L}}{\partial \mathbf{A_i}} \mathbf{Z_{i-1}^{-1}} \odot \sigma'(\mathbf{H_{i-1}}))\mathbf{Z_{i-2}} - \frac{\bar{\partial L}}{\partial \mathbf{A_{i-1}}}||^2 \tag{15}$$

For simplicity, we use $\mathcal{T}_1$ and $\mathcal{T}_2$ to represent the two terms on the right hand side of the equation above, and flatten them into column vectors, that is:

$$\mathcal{T}_1 = Flatten(\mathbf{A_i^T}(\epsilon_i \mathbf{Z_{i-1}^{-1}} \odot \sigma'(\mathbf{H_{i-1}}))\mathbf{Z_{i-2}}) \tag{16}$$

$$\mathcal{T}_2 = Flatten(\mathbf{A_i^T}(\frac{\bar{\partial L}}{\partial \mathbf{A_i}} \mathbf{Z_{i-1}^{-1}} \odot \sigma'(\mathbf{H_{i-1}}))\mathbf{Z_{i-2}} - \frac{\bar{\partial L}}{\partial \mathbf{A_{i-1}}}) \tag{17}$$

So, we have:

$$||\epsilon_\mathbf{i-1}||^2 = ||\mathcal{T}_1 + \mathcal{T}_2||^2 = ||\mathcal{T}_1||^2 + ||\mathcal{T}_2||^2 + \mathcal{T}_1^T \mathcal{T}_2 + \mathcal{T}_2^T \mathcal{T}_1 \tag{18}$$

By taking the expectation of both sides of the equation, we can derive:

$$\mathbb{E}[||\epsilon_\mathbf{i-1}||^2] = \mathbb{E}[||\mathcal{T}_1||^2] + \mathbb{E}[||\mathcal{T}_2||^2] + \mathbb{E}[\mathcal{T}_1^T \mathcal{T}_2] + \mathbb{E}[\mathcal{T}_2^T \mathcal{T}_1] \tag{19}$$

Next, we will prove that $\mathbb{E}[\mathcal{T}_1^T \mathcal{T}_2] = 0$. Assume that $\epsilon_\mathbf{i} = (\epsilon_{i,j}) \in \mathbb{R}^{d_1 \times d_2}$, where $\mathbb{E}[\epsilon_{i,j}] = 0, \forall i \in [1, d_1], j \in [1, d_2]$. We first put forth some lemmas.

*Lemma 1:* If $\mathbf{A} = (a_{i,j}) \in \mathbb{R}^{d_1 \times d_2}$, and each $a_{i,j}$ is a linear combination of $\epsilon_{p,q}$, that is: $a_{i,j} = \sum_{p,q} \alpha_{p,q}^{i,j} \epsilon_{p,q}$; $\mathbf{B} = (b_{i,j}) \in \mathbb{R}^{d_2 \times d_3}$, and each $b_{i,j}$ is independent of $\epsilon_{p,q}$; $\mathbf{C} = (c_{i,j}) \in \mathbb{R}^{d_4 \times d_1}$, and each $c_{i,j}$ is independent of $\epsilon_{p,q}$. Then, each element in $\mathbf{C} \cdot \mathbf{A}$ and $\mathbf{A} \cdot \mathbf{B}$ is also a linear combination of $\epsilon_{p,q}$.

*Proof:* For each element $d_{i,j}$ in $\mathbf{C} \cdot \mathbf{A} \in \mathbb{R}^{d_4 \times d_2}$, we have

$$
\begin{align}
d_{i,j} &= \sum_k c_{i,k} a_{k,j} = \sum_k c_{i,k} \sum_{p,q} \alpha_{p,q}^{k,j} \epsilon_{p,q} \tag{20} \\
&= \sum_k \sum_{p,q} c_{i,k} \alpha_{p,q}^{i,k} \epsilon_{p,q} = \sum_{p,q} \sum_k c_{i,k} \alpha_{p,q}^{k,j} \epsilon_{p,q} \tag{21} \\
&= \sum_{p,q} (\sum_k c_{i,k} \alpha_{p,q}^{k,j}) \epsilon_{p,q} \tag{22}
\end{align}
$$

Hence, each element in $\mathbf{C} \cdot \mathbf{A}$ can be represented as a linear combination of $\epsilon_{p,q}$. Similarly, for each element $e_{i,j}$ in $\mathbf{A} \cdot \mathbf{B} \in \mathbb{R}^{d_1 \times d_3}$, we have:

$$
\begin{align}
e_{i,j} &= \sum_k a_{i,k} b_{k,j} = \sum_k b_{k,j} \sum_{p,q} \alpha_{p,q}^{i,k} \epsilon_{p,q} \tag{23} \\
&= \sum_k \sum_{p,q} b_{k,j} \alpha_{p,q}^{i,k} \epsilon_{p,q} = \sum_{p,q} \sum_k b_{k,j} \alpha_{p,q}^{i,k} \epsilon_{p,q} \tag{24} \\
&= \sum_{p,q} (\sum_k b_{k,j} \alpha_{p,q}^{i,k}) \epsilon_{p,q} \tag{25}
\end{align}
$$

This finishes the proof for Lemma 1.

*Lemma 2:* If $\mathbf{A} = (a_{i,j}) \in \mathbb{R}^{d_1 \times d_2}$, and $a_{i,j} = \sum_{p,q} \alpha_{p,q}^{i,j} \epsilon_{p,q}$; $\mathbf{B} = (b_{i,j}) \in \mathbb{R}^{d_1 \times d_2}$, and each $b_{i,j}$ is independent of $\epsilon_{p,q}$. Then, each element in $\mathbf{A} \odot \mathbf{B}$ is also a linear combination of $\epsilon_{p,q}$.

*Proof:* For each element $f_{i,j}$ in $\mathbf{A} \odot \mathbf{B} \in \mathbb{R}^{d_1 \times d_2}$, we have:

$$
f_{i,j} = a_{i,j} b_{i,j} = b_{i,j} \sum_{p,q} \alpha_{p,q}^{i,j} \epsilon_{p,q} = \sum_{p,q} (b_{i,j} \alpha_{p,q}^{i,j}) \epsilon_{p,q} \tag{26}
$$

Based on Lemma 1 and Lemma 2 above, we can conclude that $\mathcal{T}_1^T \mathcal{T}_2$ can be expressed as a linear combination of $\epsilon_{p,q}$, that is:

$$
\mathcal{T}_1^T \mathcal{T}_2 = \sum_{p,q} \mathcal{C}_{p,q} \epsilon_{p,q} \tag{27}
$$

where each $\mathcal{C}_{p,q}$ does not contain any term associated with $\epsilon_{p,q}$. According to Assumption 2, the random variable $\mathcal{C}_{p,q}$ is independent of $\epsilon_{p,q}$, so we can simplify its expectation form as:

$$
\mathbb{E}[\mathcal{T}_1^T \mathcal{T}_2] = \mathbb{E}[\sum_{p,q} \mathcal{C}_{p,q} \epsilon_{p,q}] = \sum_{p,q} \mathbb{E}[\mathcal{C}_{p,q} \epsilon_{p,q}] = \sum_{p,q} \mathbb{E}[\mathcal{C}_{p,q}] \mathbb{E}[\epsilon_{p,q}] = \sum_{p,q} \mathbb{E}[\mathcal{C}_{p,q}] \cdot 0 = 0 \tag{28}
$$

By the same reasoning, we can conclude that $\mathbb{E}[\mathcal{T}_2^T \mathcal{T}_1] = 0$. Finally, we have:

$$
\begin{align}
\mathbb{E}[||\epsilon_{\mathbf{i-1}}||^2] &= \mathbb{E}[||\mathcal{T}_1||^2] + \mathbb{E}[||\mathcal{T}_2||^2] + \mathbb{E}[\mathcal{T}_1^T \mathcal{T}_2] + \mathbb{E}[\mathcal{T}_2^T \mathcal{T}_1] \tag{29} \\
&= \mathbb{E}[||\mathcal{T}_1||^2] + \mathbb{E}[||\mathcal{T}_2||^2] \tag{30}
\end{align}
$$

According to Assumption 1, $\mathbb{E}[||\mathcal{T}_1||^2] = \mathbb{E}[||\epsilon_{\mathbf{i}}||^2]$, so:

$$
\begin{align}
\mathbb{E}[||\epsilon_{\mathbf{i-1}}||^2] - \mathbb{E}[||\epsilon_{\mathbf{i}}||^2] &= \mathbb{E}[||\mathcal{T}_1||^2] - \mathbb{E}[||\epsilon_{\mathbf{i}}||^2] + \mathbb{E}[||\mathcal{T}_2||^2] \tag{31} \\
&= \mathbb{E}[||\mathcal{T}_2||^2] \geq 0 \tag{32}
\end{align}
$$

This finishes the proof of Theorem 1.

## 6.2 PROVING DIVERGENCE ACCUMULATION WITH OTHER ARCHITECTURES

### 6.2.1 EXTENSION FOR CNN

First and foremost, it is crucial to clarify that the convolution operation is fundamentally a linear operation. As a result, the convolution layer can be seen as a special type of linear layer to some extent.

However, it possesses unique properties: 1) Local connectivity. Unlike the connections in linear layers which are dense, only pixels in the same neighborhood can participate in the computation; 2) Parameter sharing. This means that we use the same convolution kernel for all parts of the image. This implies that a convolution layer can be transformed into a matrix multiplication form similar to the linear layer.

Next, back to our proof. The key that makes the proof for CNNs different lies in parameter sharing. Parameter sharing results in the gradient of each parameter being accumulated from multiple computed gradients during backpropagation. However, if we assume that the gradients of each parameter within the same layer are independent, we can still guarantee the correctness of the proof. Without loss of generality, let's assume there are two layers, each with two parameters, namely $[w_{1,1}, w_{1,2}]$, $[w_{2,1}, w_{2,2}]$, and their gradient errors are $[\epsilon_{1,1}, \epsilon_{1,2}]$, $[\epsilon_{2,1}, \epsilon_{2,2}]$ respectively. We have not yet introduced parameter sharing here, so according to theorem 1 in our paper, we have:

$$\mathbf{E}[||(\epsilon_{1,1}, \epsilon_{1,2})||_2^2] \geq \mathbf{E}[||(\epsilon_{2,1}, \epsilon_{2,2})||_2^2] \quad (1)$$

When introducing parameter sharing, and assuming the parameters within each layer are shared, the expected gradient norm for the first layer is:

$$\mathbf{E}[||(\epsilon_{1,1} + \epsilon_{1,2})||_2^2] = \mathbf{E}[||\epsilon_{1,1}||_2^2] + \mathbf{E}[||\epsilon_{1,2}||_2^2] = \mathbf{E}[||(\epsilon_{1,1}, \epsilon_{1,2})||_2^2]$$

Similarly,

$$\mathbf{E}[||(\epsilon_{2,1} + \epsilon_{2,2})||_2^2] = \mathbf{E}[||(\epsilon_{2,1}, \epsilon_{2,2})||_2^2]$$

Substituting into equation (1), we can see that divergence accumulation still holds. In this way, we have extended the proof to the realm of CNNs.

### 6.2.2 EXTENSION FOR SKIP CONNECTION:

First, let's agree on some basic notations. Assume $\mathbf{X}$ is the intermediate result from the output of the starting point of a skip connection. Due to residual link, we denote the $\mathbf{X_1}$ as $\mathbf{X}$ that directly sent to the next layer, and denote $\mathbf{X_2}$ as $\mathbf{X}$ sent to the later layers via skip connections.

Next, let the gradients of $\mathbf{X}$, $\mathbf{X_1}$ and $\mathbf{X_2}$ to be:

$$\frac{\partial L}{\partial \mathbf{X_1}} = \frac{\bar{\partial L}}{\partial \mathbf{X_1}} + \epsilon_1, \frac{\partial L}{\partial \mathbf{X_2}} = \frac{\bar{\partial L}}{\partial \mathbf{X_2}} + \epsilon_2, \frac{\partial L}{\partial \mathbf{X}} = \frac{\bar{\partial L}}{\partial \mathbf{X}} + \epsilon$$

Given that $\mathbf{X_1}$ and $\mathbf{X_2}$ are two computation paths, so the gradient of $\mathbf{X}$ is actually their sum. Hence, we have:

$$\epsilon = \epsilon_1 + \epsilon_2$$

If we further assume that $\epsilon_1$ and $\epsilon_2$ are independent, we have:

$$||\epsilon||_2^2 = ||\epsilon_1||_2^2 + 2{\epsilon_1}^T \epsilon_2 + ||\epsilon_2||_2^2 = ||\epsilon_1||_2^2 + ||\epsilon_2||_2^2 \geq \max\{||\epsilon_1||_2^2, ||\epsilon_2||_2^2\}$$

The reason for the second equation is the same as Eq. (28) in our supplementary materials. So far, we have proven that after adding skip connections, the output value of the previous layer has a larger $\epsilon$, thus our divergence accumulation still holds.

## 6.3 ADDITIONAL EXPERIMENTS

### 6.3.1 DETAILED LAYER-WISE DIVERGENCES

In Section 4, we demonstrate the Mean Divergence (Div.) under different experimental settings, but without showing the differences among different layers. Here, we provide more detailed experimental results for divergences across different layers.

The experiment settings in the following figures are the same as those in Section 4. Each figure caption indicates the dataset used and the number of layer divergence plotted.

Fig. 4 illustrates divergences using different data-augmentation techniques. It is evident that by applying appropriate data-augmentation methods, we can obtain lower and more stable model divergences, ultimately enhancing performance.

Fig. 5 shows model divergences using different input image resolutions. We observe that when a smaller resolution is used, the model divergences for deep layers are significantly larger. This is because deeper layers tend to have larger receptive fields, and if the resolution is not sufficient, the receptive field of deep layers may cover the entire image, leading to greater data dissimilarity and larger divergences.

Fig. 6 presents model divergences using different model widths. It is clear that a wider model architecture results in a significant decline in divergence. This aligns with our expectation that wider models possess the "lazy" property, updating parameters more mildly.

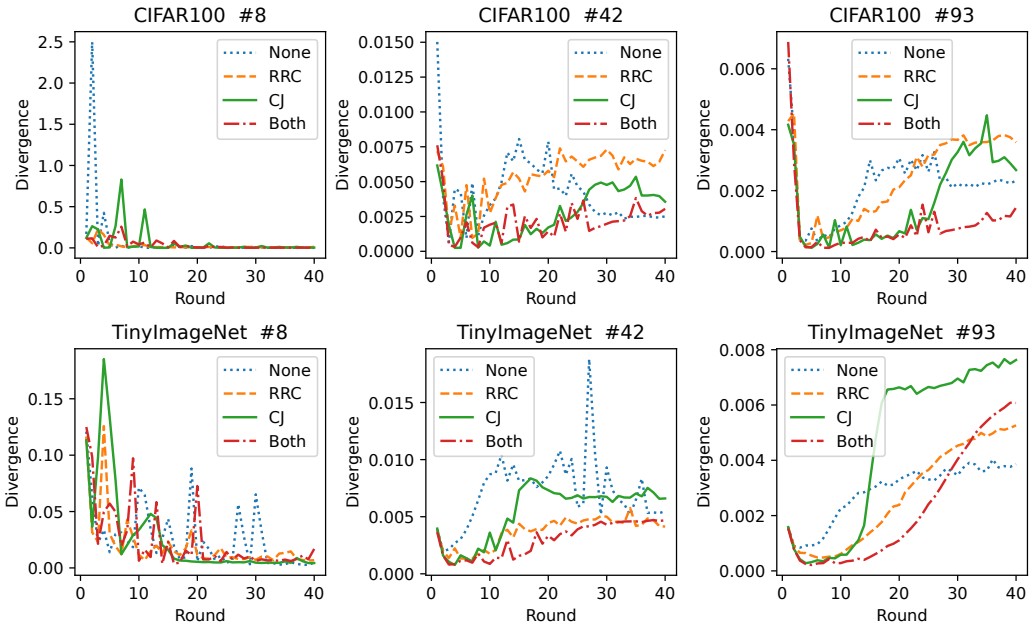

Figure 4: Divergences for different data-augmentation methods.

### 6.3.2 EFFECT OF RESIDUAL LINK

Residual link is a classic method in CL, which can effectively improve the performance of deeper neural networks. In FL, as analyzed in Section 3, divergence accumulates and amplifies during the model's back-propagation process. Residual connections can provide a "shortcut" for back-propagation, allowing gradients in deep layers to propagate quickly to shallower layers. As a result, since the length of the back-propagation path is reduced, the accumulation of model divergence is also reduced, and the convergence speed is faster. As can be seen from the experimental results in Table 6.3.2 although there is no significant difference in the final convergence accuracy of federated learning before and after adding residual connections, the convergence speed has increased.

### 6.3.3 AVOID USING DECREASED CHANNEL DIMENSIONS

In this part, we include an additional experiment. As mentioned in the introduction, we observe that the divergence of shallow layers tends to decrease and eventually converge, whereas the divergence of deep layers tends to intensify.

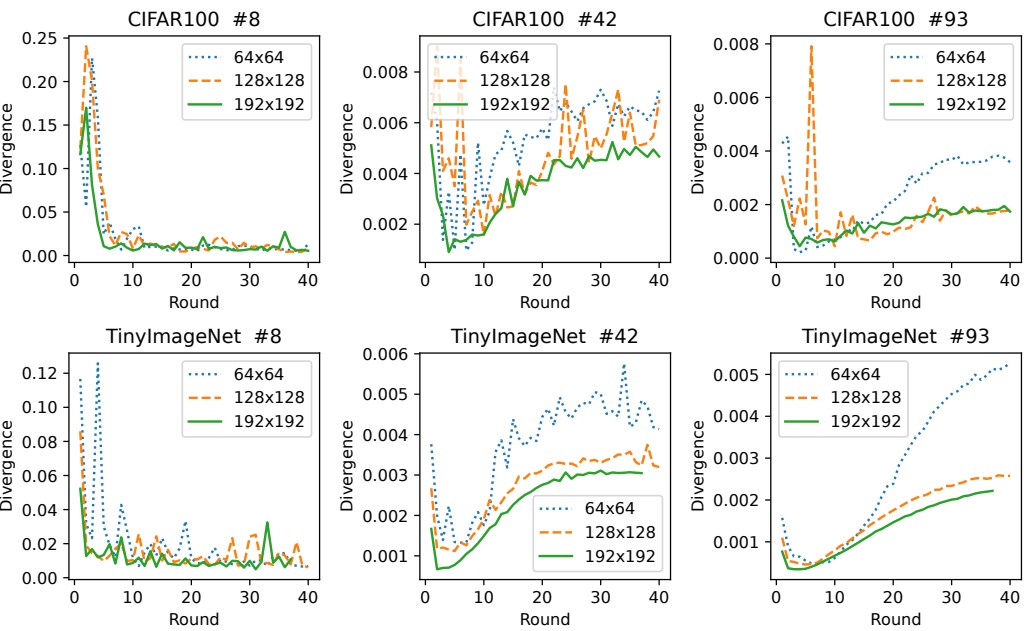

Figure 5: Divergences for different image resolutions.

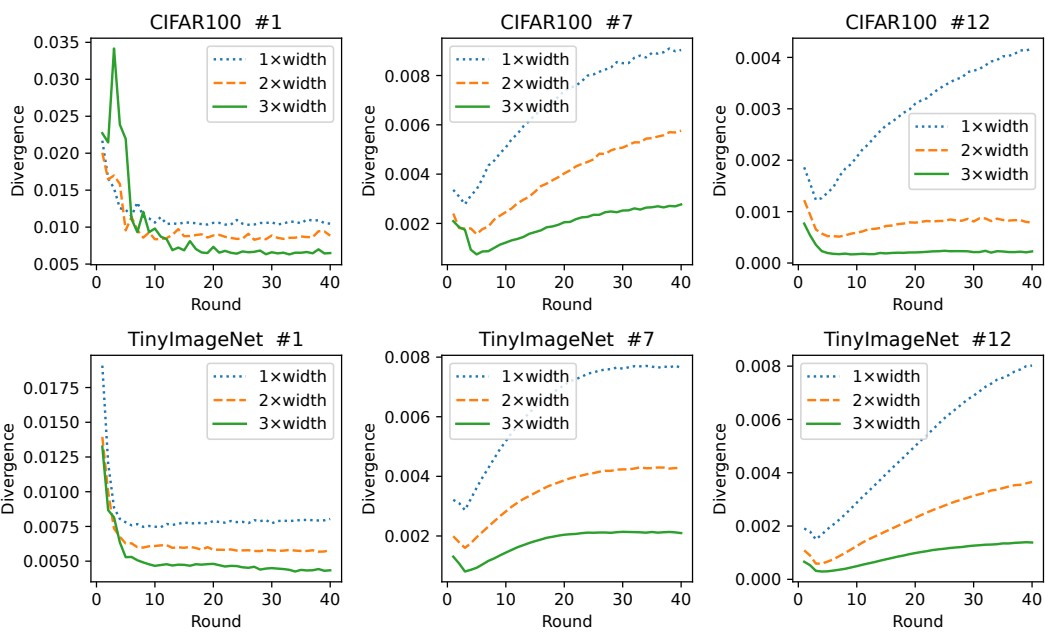

Figure 6: Divergences for different model width.

Intuitively, in order to leverage this property, we can assign a more significant role to the shallow layers compared to the deep layers. Conventionally, the number of channels in a CNN increases gradually as the layers go deeper. However, our proposed approach follows an opposite design principle where the number of channels gradually decreases.

Table 6: We show different results with residual link and without residual link. Acc. is the best model accuracy at test time and #Rnd. is the number of communication round when test accuracy first exceeds 0.4, 0.50.

| Dataset | Method | With Res. Link | | No Res. Link | |
|---------|--------|------|------|------|------|
| | | Acc. | #Rnd. | Acc. | #Rnd. |
| Tiny ImageNet | CL | 0.509 | - | 0.522 | - |
| | FL | 0.458 | 21 | 0.454 | 24 |
| CIFAR100 | CL | 0.587 | - | 0.569 | - |
| | FL | 0.527 | 30 | 0.526 | 33 |

For the experiment setup, we selected three models for comparison: a model with channel dimensions of (32, 64, 128, 256) following the typical design, a model with channel dimensions of (120, 120, 120, 120) referred to as the Mean design, and a model with channel dimensions of (256, 128, 64, 32) denoted as the Reversed design. In the subsequent results analysis, we refer to these designs as Normal, Mean, and Reversed, respectively.

Surprisingly, we discovered that the Normal design, which adheres to the typical channel dimension pattern, exhibited the best performance with the lowest divergence. The final accuracies achieved by the three designs were 45.8%, 44.0%, and 35.4%, respectively. To visualize the divergences across different layers, we plotted them layer-wise in Fig. 7.

As depicted in the figure, the Reversed network displayed consistently larger divergences in the deep layers compared to the Normal network. Additionally, the divergences in the shallow layers of the Reversed network continuously intensified without showing any signs of convergence. This behavior can be attributed to the smaller width of the deeper layers, causing their divergences to be initially large due to the "lazy" property discussed in Section 4. Consequently, these large divergences accumulate onto the divergences of the shallow layers, resulting in a convergence issue.

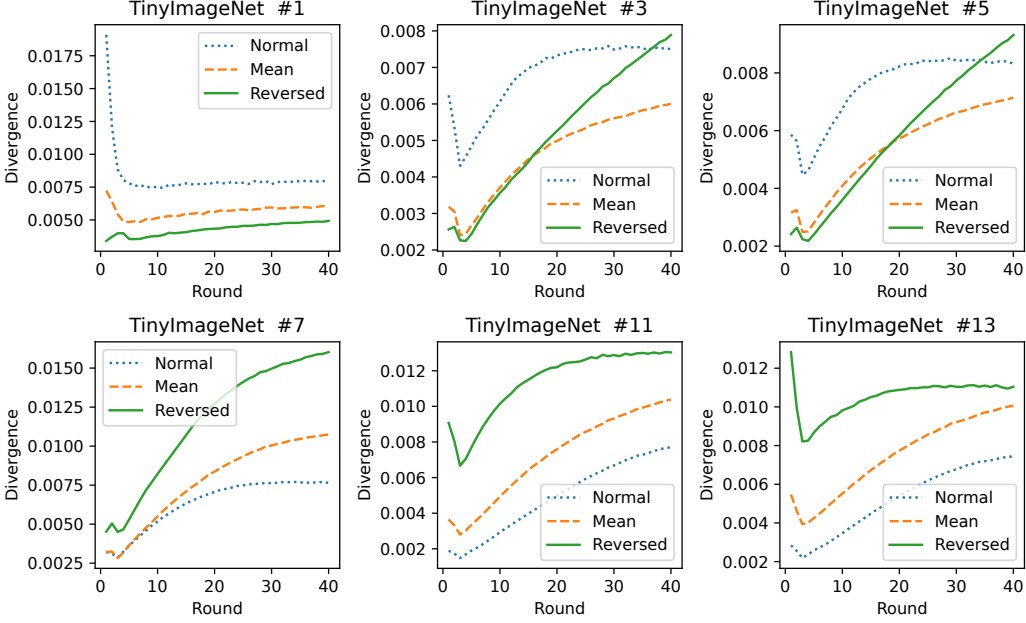

Figure 7: Divergences for different model architectures.