# OpenReview forum: "Unlocking the Potential of Federated Learning for Deeper Models"
_ICLR.cc/2024/Conference — ICLR 2024 Conference Withdrawn Submission_

### Official Review · Reviewer_wvyK · 2023-10-19

**Soundness:** 1 poor
**Presentation:** 2 fair
**Contribution:** 1 poor
**Rating:** 1
**Confidence:** 4

**Summary:**

Federated learning has been mainly utilized for relatively shallow neural nets. The authors show that increasing depth can significantly worsen performance even for iid client data. However, they also found--perhaps counter-intuitively--that increasing width improves performance, so they argue that overfitting is not the reason. They attribute this degradation in performance to the accumulation of divergent terms in back propagation due to client dissimilarities.

**Strengths:**

- The authors give guidelines on how to improve models in federated learning.
- They run various experiments and try to prove the divergence accumulation phenomenon.

**Weaknesses:**

Various claims and steps in this paper are flawed. For example, it is not surprising that increasing width increases performance. The problem is that increasing depth is decreasing performance in your case, which is something that need to be investigated in depth and demonstrated with careful experiments because it is against the strongest point of "deep" learning and its common wisdom.

Another thing is that federated learning can be reduced to SGD in the simplest setting (one local step, iid clients, etc.), so theoretically, the divergence that might have been introduced could be from local training for more than one step. However, this has never been discussed in the paper, except the fact that the local epochs are set to 8 (which is a lot by the way). Finally, the authors explicitly mention client dissimilarity, yet they assume iid clients, which makes the authors objectives from this paper slightly unclear.

Here are some other comments:
1. The authors assume that the clients data are identically distributed, but proceed to show that they are not. Equation (6) assumes that the difference between clients data is a linear term, which is simply not often the case for non-iid clients. Most data-augmentations are non-linear, for example.
2. In assumption 1 and 2, the stochasticity of the difference of gradients comes from the data distribution, so saying that the error is independent from the intermediate calculations Z and H--from which the error itself is eventually calculated--is very hard to believe.
3. Assumptions and Theorems should be formatted correctly.
4. Observations 1 and 2 are the same.
5. Divergence measure in Fig. 3 is not clear. What does it stand for? Which direction is better?
6. Modeling data diversity with a Gaussian noise is simply wrong. In fact, Gaussian noise is sometimes added to the data to improve model's stability/generalization.
7. The authors did not use any of the well-known federated datasets (e.g. LEAF datasets) and did not specify how the data is partitioned into the clients.
8. The conducted experiments are the same with different models. The authors need to provide more detailed experiments in order to demonstrate that this phenomenon is truly caused by the depth of neural nets (see questions below).
9. Lemma 1 is simply saying that a linear function a linear combination of epsilon produces a different linear combination of epsilon, which is trivial. No need for independence as well. Same thing with Lemma 2.

**Questions:**

- Did you use batch normalization? This is very important.
- What optimizer did you use?
- Can you run an experiment that demonstrates increasing divergence as you increase depth, and the relationship between divergence and depth for a set of fixed widths? Use a simple net like: linear -> relu -> linear -> relu -> ... and make sure to normalize your divergence measure with respect to the depth.
- Can you share anonymous source code?

---

### Official Review · Reviewer_pCdo · 2023-10-21

**Soundness:** 2 fair
**Presentation:** 3 good
**Contribution:** 2 fair
**Rating:** 6
**Confidence:** 2

**Summary:**

The authors observe that deeper models in the context of FL often face challenges in achieving convergence, resulting in a degradation of performance. They provide technical guidelines for improving the performance of FL on deeper models and demonstrate through experiments that these methods have a remarkable impact in reducing divergence, resulting in significantly greater enhancements in FL performance compared to centralized learning.

**Strengths:**

The paper is well-written and easy to follow. The paper focuses specifically on the challenge of applying federated learning to deeper neural networks, which is an important problem.

**Weaknesses:**

The analysis of divergence accumulation is primarily based on a simplified linear layer with an activation function. However, since the authors conducted experiments using ResNet, it would be more appropriate for them to provide the analysis based on the residual module. Additionally, the process of deriving the entire formula lacks clarity and is challenging to comprehend.

**Questions:**

1. Why does the accuracy of CL also decrease as the model becomes deeper in Figure 1?

2. Please provide a more detailed explanation of Equation (6) and clarify the meaning of $\epsilon_i$. Additionally, explain why $\epsilon_i$ is defined as shown below Equation (10). There is a typographical error in Theorem 1.

3. How was Assumption 1 derived, and why is $E[\epsilon_i]=0$?

4. Neural Architecture Search (NAS) [1] could be a more suitable tool to replace the proposed guidelines as it can automatically adjust the width and depth of the model. Inspired by NAS, several works have introduced advanced Auto Data Augmentation (ADA) methods, such as AutoAugment [2], to automatically search for data augmentation policies for different tasks. The authors should further discuss these methods for future research.

[1] He X, Zhao K, Chu X. AutoML: A survey of the state-of-the-art[J]. Knowledge-Based Systems, 2021, 212: 106622.
[2] Cubuk E D, Zoph B, Mane D, et al. Autoaugment: Learning augmentation policies from data[J]. arXiv preprint arXiv:1805.09501, 2018.

**Details Of Ethics Concerns:**

null

---

### Official Review · Reviewer_w79q · 2023-10-26

**Soundness:** 2 fair
**Presentation:** 3 good
**Contribution:** 3 good
**Rating:** 3
**Confidence:** 4

**Summary:**

This paper investigates the model depth in federated learning and identifies 'divergence accumulation', supported by both theoretical derivations and empirical evidence. The authors propose several technical guidelines and perform evaluations on three public datasets to show their effectiveness.

**Strengths:**

- This paper investigates an interesting topic.
- The introduction well presents the motivation.
- The overall study design is easy to follow.
- The authors try to provide both theoretical and empirical evaluations on the divergence accumulation.

**Weaknesses:**

- Observations in Fig 2(a) may not be accurate. divergence of deep layers also tends to increase and converge if the model is trained more than 40 rounds.
- Given \epsilon_i with Z, the assumption 2 that assumes \epsilon_i is dependent with Z and H may not hold.
- The theorem of divergence accumulation only proves linear layers. It has no consideration for other important layers in modern deep neural networks,e.g., convolutional layers and normalization layers.
- The proposed theorem does not show specific properties related to FL. It is also reasonable for centralized learning.
- The empirical evaluation is too toy; an 8-layer CNN is not a ‘deep model’. Evaluations using a real deep model (e.g., networks with overall 100 layers) are expected.
- The experimental design cannot fully validate the uniqueness of divergence accumulation in FL. Both CL and FL suffer from divergence accumulation.
- Some experiment details are not clear, E.g., the total training rounds, and data split. Given the observation in Fig.2 (a), models may not converge given only 40 rounds, and this may lead to a wrong observation.
- The experiment settings are not consistent. Tables 1 and 2 use different backbones, which is weird.
- The observation of using smaller receptive fields may not be true given the original input size is small (64x64).

**Questions:**

- Local data within each client is small. It is easy to imagine that local training with limited data cannot present better performance than centralized training. What are the results if we increase the local client data to the same amount as the centralized one? If this observation still exists, it means the divergence would be specifically related to the FL training paradigm rather than short of data.
- In Fig.2(a), layer #23 and #42 show similar divergence, which may not well fit the divergence accumulation theorem. Any explanations?